# Risk Factors Associated with Musculoskeletal Injuries within the Crew of the Leopard 2 A6 Main Battle Tank Using Inertial Movement Unit Sensors: A Pilot Study

**DOI:** 10.3390/s24144527

**Published:** 2024-07-12

**Authors:** Bruno Pedro, Ana Assunção, Filomena Carnide, Beatriz Damião, Rui Lucena, Nuno Almeida, Paula Simões, António P. Veloso

**Affiliations:** 1Laboratório de Biomecânica e Morfologia Funcional, CIPER, Faculdade de Motricidade Humana, Universidade de Lisboa, 1499-002 Cruz Quebrada-Dafundo, Portugal; bmpedro@fmh.ulisboa.pt (B.P.); aassuncao@edu.ulisboa.pt (A.A.); fcarnide@fmh.ulisboa.pt (F.C.); 2Centro de Investigação Desenvolvimento e Inovação da Academia Militar (CINAMIL), Instituto Universitário Militar, Academia Militar, 1169-203 Lisbon, Portugal; damiao.bc@exercito.pt (B.D.); rui.lucena@academiamilitar.pt (R.L.); almeida.nrc@academiamilitar.pt (N.A.); paula.simoes@academiamilitar.pt (P.S.); 3MRLab—Military Readiness Lab, 2720-113 Amadora, Portugal; 4CMA, Faculdade de Ciências e Tecnologia, Universidade Nova de Lisboa, 1099-085 Lisbon, Portugal

**Keywords:** risk factors, musculoskeletal disorders, main battle tank Leopard 2 A6, REBA, level of risk, inertial measurement units

## Abstract

This study aims to assess the musculoskeletal risk of military personnel on a Leopard 2 A6 main battle tank crew and to identify associated factors for future prevention and mitigation strategies. A sample of 57 Portuguese military personnel, who are or were part of the Leopard 2 A6 main battle tank crew, answered a questionnaire on their perception of task performance, considering muscle demands, comfort, posture, movements, and associated symptoms. A subsample of four soldiers from the Armoured Squadron of the Portuguese Mechanized Brigade were assessed using an inertial measurement unit system and underwent a whole-body kinematic analysis coupled with a Rapid Entire Body Assessment during a simulated two-hour mission. The results indicate that soldiers accurately perceive their roles within the crew and that, overall, there is a high risk of musculoskeletal injuries in all tasks. However, tasks directly related to the crew’s primary duties carry consistently high risk when considering the time spent on their tasks. This study highlights the need for targeted preventive measures to reduce the incidence and severity of injuries among the crew of the Leopard 2 A6 main battle tank.

## 1. Introduction

Musculoskeletal injuries (MSKIs) are the leading cause of medical discharges and medical downgrades in the armed forces. They have a major impact on deployability and combat effectiveness and are endemic in the military population [1]. Previous military studies have shown that 30–70% of military injuries are musculoskeletal overuse injuries (MSOIs) [2,3]. Nevertheless, most of these studies and national surveillance systems only include the diagnosis of acute traumatic injuries [4,5]. However, many MSKIs result from cumulative microtraumatic mechanisms, and the burden of these on healthcare systems, communities, and workplaces is often underestimated. The majority of MSKIs affect the spine and lower extremities, with a prevalence of approximately 40%, including sprains, strains, and tendinopathies [6]. Inflammation and pain are responsible for 82% of all types of injuries. Broken down further, lower back injuries account for 48.5% of all spine injuries, and knee and lower leg injuries account for 57.3% of all lower limb injuries [7].

The main risk factors for MSKI are classified as non-modifiable factors (e.g., female gender, older age groups, and previous injuries) or modifiable factors because their frequency can be reduced by intervention programmes (e.g., body mass index, movement patterns, carrying heavy loads, exposure to vibration, prolonged standing, and long walking) [8]. These modifiable factors are present in both higher-training-load exercise situations and in ordinary routine operations in a military context. The difference is that military personnel operate for longer periods of time and with other uncontrolled variables (e.g., weather, terrain conditions), so these injuries typically last five times longer than traumatic MSKIs [9,10]. The low physical fitness of military personnel during vigorous activity has also been identified as a possible cause of MSKIs. However, there are no conclusive data to support this possibility. For these reasons, it is not enough to know the popular areas in which injuries are bound to occur. It is also important to know what activity was being performed at the time of the injury and how much time was spent on those activities. 

Among MSKIs, leading tasks are running (43%), work-related tasks (11%), falling (10%), road machinery (8%), and sports (7%) [11]. However, when it comes to risk factors, there are studies that focus not on the activity itself but on the time spent on those activities [12]. Therefore, special attention should be paid to occupational exposure patterns, namely the frequency, intensity, and duration of each risk factor [13]. Whether physical factors (mechanical exposure) or work-related psychosocial factors are involved, there is no perfect method for identifying and measuring risk factors. Each method has advantages and disadvantages, and the decision must be made based on the work context and activity under study, the level of accuracy and precision required, the quality of results required, the available costs and resources available, and the variety of factors, which must be measured at the same time. 

From direct quantitative measurements to observational methods and to subjective assessment techniques, there is a wide range of available methods. As we improve the quality of results and the accuracy of analysis, the costs of time and expertise required to analyse the data also increase. [14,15,16]. However, due to the nature of military soldiers’ activity, quantitative measurements should be prioritized as normative values for other occupational settings may not be consistent with military exposure. 

There is little research addressing the incidence of MSKI among MBT crews. A study comparing the Danish Army’s MBT crew with personnel from other unit types found that the MSKI pattern did not differ significantly between different units [17]. Within the MBT crew, working as a gunner for less than two years was associated with an increased risk of neck pain, while working as a loader was associated with a higher risk of shoulder pain [17]. Another study among German MBT commanders found significant back and knee pain after a 320 km road deployment [18]. Therefore, there is a need to develop effective prevention strategies for MSKIs by identifying the causes and risk factors [19]. One of these strategies is the assessment of the risk factors presented, which is part of the risk management process and contributes to the analysis of accidents and potential health threats, with the aim of reducing possible work-related risks by identifying them. Once these risks are identified, measures should be taken to reduce and/or minimize their occurrence [15]. Measuring the exposure to factors contributing to MSKIs serves as a basis for prevention and risk reduction measures. 

However, as far as these strategies are concerned, MSKI prevention in military settings is underdeveloped and is identified and considered a priority research topic [17,19]. Thus, there is an existing gap in the research on MSKI in the Portuguese Army caused by tasks with a high physical demand or repetitive execution, as well as in research on the prevention of MSKI. Thus, the aim of this study was to determine the level of musculoskeletal risk among a Leopard 2 A6 MBT crew in the performance of their duties and to identify associated risk factors with the aim of developing mitigation measures for these specific military personnel to reduce the incidence of these injuries and their possible consequences.

## 2. Materials and Methods

### 2.1. Study Design

This was a single experimental study (simulation) conducted with Portuguese Army military personnel. 

### 2.2. Participants

The population of 57 military personnel were asked to respond to a questionnaire. The inclusion criteria were both genders; at least one year of experience in military service; military personnel performing or having participated in the duties of tank commander, gunner, driver, and loader. The exclusion criterion was that an MSKI was present one year before data collection. A subgroup of four military personnel was randomly selected to perform the main daily tasks, to collect quantitative data on the biomechanical workload. Before starting the data collection, all participants were informed about this study and its aims and procedures, and an informed consent form was read and signed.

### 2.3. Musculoskeletal Pain Intensity

The military personnel were asked to report their discomfort and pain in 15 body regions using a Numeric Pain Rating Scale (NPRS) [20]. The NPRS is a self-report instrument used in this study to assess pain status. The scale was presented with numbers from 0 to 10 with the following anchors at each end of the scale: “no pain” and “worst possible pain”. The NPRS has been shown to be reliable with an intraclass correlation coefficient (ICC) of 0.76 (95% confidence interval: 0.51–0.87) [20]. The participants reported their pain at two different times: during task performance and after task performance.

### 2.4. MBT Tasks—Experimental Protocol

Before data collection the four military personnel, the tank commander, the gunner, the driver, and the loader had the opportunity to familiarize themselves with the respective sequence of tasks of the experimental protocol in the Leopard 2 A6 MBT (Table 1, Table 2, Table 3 and Table 4) until they felt comfortable with performing the movements without assistance. Kinematic analysis and risk assessment were then carried out in the four tasks of a Leopard 2 MBT. The sequence of tasks was analysed based on a two-hour deployment and allowed tasks to be divided according to execution time. 

### 2.5. Instrumentation

Whole-body kinematic data were recorded at 240 Hz with 17 IMU sensors (Xsens MVN Technologies, Enschede, The Netherlands) [21], using the Xsens MVN Analyse software (version 2019.2). The Xsens MVNV Link system consists of 17 IMUs (36 × 24 × 10 mm, 10 g), each containing 3D gyroscopes, 3D accelerometers, and a magnetometer, connected to a case and a battery. These sensors were placed on both the left and right body segments of a Lycra suit after the participant put it on, over the feet, shanks, thighs, scapulae, upper arms, forearms, and hands, one sensor in the sternum, one in the head, and one in the pelvis (Figure 1A,B). Following the manufacturer’s instructions and, if possible, positioning over the bone to reduce soft tissue artefacts [22]. In addition, the body, battery, and cables were housed in the Lycra suit [23,24]. After all the sensors were placed, the following anthropometric measurements were collected from each participant to scale their avatar model: standing height, shoe length, arm span, ankle, knee, hip and shoulder heights, hip and shoulder width, and shoe sole height [21,23] (Figure 1A). A calibration was then carried out to align the sensors to the respective segment [23]. This calibration was performed by having participants stand in a neutral “n-pose” position with the palm facing medially, then walk forward a few meters, return to the starting position, and stand in “n-pose” again [23] (Figure 1C). Following the calibration procedures, each military personnel individually performed a set of tasks inherent to their role in the MBT (Table 1, Table 2, Table 3 and Table 4).

### 2.6. Kinematic Data Processing

After reprocessing in HD quality, the motion files obtained with the IMU system were exported from the Xsens MVN Analyse software in MVNX format and opened in the motion analysis software Visual 3D (V6, C-motion, Inc., Germantown, ML, USA). Using the position and orientation information (POSE) of the segments contained in the MVNX file, the joint angles of the neck, trunk, shoulder, elbow, wrist, and knees were calculated using a ML–AP–Axial Cardan sequence [25]. The POSE of the segments was calculated by the MVN Fusion Engine, based on a biomechanical model created in the Xsens MVN Analyse software during the calibration procedure that preceded the collection of the tasks. The origin, dimensions, and anatomical axis definitions of the segments followed the manufacturer’s recommendations, as explained in detail in the Xsens MVN user manual [24]. Representative Positions of the Crew during the Performed Tasks can be seen in Appendix A.

### 2.7. Assessment Risk

To determine the biomechanical risk associated with the performance of each task, joint angle data were obtained and used to determine the Rapid Entire Body Assessment (REBA) scores method [26]. Due to the continuous data collection during task performance, the most challenging posture was chosen for each step of the sequence of each task. The angles were determined using a reference frame (x, y, z), and based on the planes and basic movements, the movements performed were assigned to three angles provided by the Visual 3D software in each task. The REBA method can provide an overview of the risk of MSKIs using predefined cutoff values. REBA classifies scores into negligible risk (1), low risk (2 to 3), medium risk (4 to 7), high risk (8 to 10), and very high risk (more than 11).

### 2.8. Normalized REBA Scores

After calculating the risk level of tasks using the REBA method, the scores were adjusted for the time spent in each step of the sequence to obtain a final risk score. The risk score (SR) was determined based on Equation (1).
(1)$${SR}_{i}={REBA}_{i}\times \frac{T_{i}\times 100}{T_{Mission}},\;(i=1,2,\ldots ,15)$$
where REBA_i_ is the final REBA score, i is the analysed task, T_i_ is the execution time of the task, and T_Mission_ is the total duration of the mission.

To compare the risk scores of tasks inherent in the four functions of the CC crew, an indicator was calculated, the normalized risk score (SRn), given by Equation (2).
(2)$${SRn}_{i}=\frac{{SR}_{i}}{{SR}_{Max}}$$
where SR_i_ is the calculated risk score and SR_Max_ is the highest risk score value.

As can be seen in Table 5, this indicator allows the standardization of the risk score values between 1 and 100, where these intervals refer to the values of the REBA risk levels and their time duration.

It was, therefore, possible to classify the task performed by each of the four military members of the crew based on their normalized risk score and the possible actions related to that risk.

### 2.9. Statistical Analyses

A descriptive analysis was performed using means and standard deviations and percentiles for quantitative variables and absolute and relative frequencies for qualitative o variables. The data obtained from REBA assessment and pain intensity were tested for normality (Shapiro–Wilk test) and homogeneity of variances (Levene test). If both conditions were not observed, a one-way ANOVA, with Welsh correlation, was conducted to analyse the differences between the military tasks. The post hoc Tamhane method was used to determine which REBA scores differed between the MBT tasks. Spearman correlation between the REBA scores and the pain intensity was then performed to analyse the influence of biomechanical exposure on the experienced symptoms. The same applies here: if the normality of the variables is not observed, we use the Spearman correlation. Statistical analysis was performed using IBM SPSS Statistics version 28.0 (SPSS Inc., an IBM company, Chicago, IL, USA). The significance level was set to *p* < 0.05. 

## 3. Results

### 3.1. Sociodemographic Characteristics of the Participants

The descriptive statistics of the larger sample are presented in Table 6. Of the total sample, 12.3% were women. The highest mean age was 33.65 ± 6.71 years in the position of tank commander.

The characteristics of the subsample of military personnel who performed the kinematic analysis are presented in Table 7. 

### 3.2. Musculoskeletal Pain

Figure 2 shows the pain intensity reported by each military personnel while performing the task. The loader was the one who reported the lowest pain intensity in all segments, with the knees being the body region with the highest intensity. The low back was the area of greatest pain intensity reported by the commander, gunner, and driver. In these three tasks, the back region was the one that showed the highest intensities.

Figure 3 shows the intensity reported by each military member on completing the task. As with the pain intensity reported during the task, the low back was the segment with the highest pain intensity in all tasks, including the loader. 

### 3.3. Kinematic Data and Assessment Risk Analysis

Figure 4 shows the REBA scores for the task sequences of all tasks. For the four tasks, most steps in the sequence are classified as high risk (marked in red). However, there are some steps that are considered medium risk. It should be noted that at Step 10 (placing the cartridge into the cartridge chamber) of the loader sequence, the REBA score was at its highest, meaning urgent intervention is required.

The normalized score for each task sequence for each position is presented in Table 8. As expected, the tasks performed over a long period were of time were rated between high and very high risk. For the gunner, the driver, and the tank commander, there is only one task, which is considered very high risk and is carried out over a long period of time. 

Regarding the biomechanical exposure analysis, there is a statistically significant difference between REBA normalized between tank commander and gunner tasks (mean difference = 8.55; *p* < 0.001), as well as between gunner and driver tasks (mean difference = 1.50; *p* < 0.001) and the loader task (mean difference = 1.5; *p* < 0.001). However, when looking at the crude REBA score, there were no statistical differences between all tasks. These results call attention to the need to consider the exposure time for each biomechanical risk factor, to account for the daily cumulative effect of that exposure when performing the task. 

No statistically significant differences were found between crude REBA scores and pain intensity in different body regions either during task performance or afterwards. 

## 4. Discussion

Within the military population, MSKIs continue to represent a major concern due to their prevalence and impact on operational readiness. The aim of this study was to assess the level of musculoskeletal risk experienced by members of the crew of the Leopard 2 A6 MBT during their deployment, with the aim of identifying the contributing factors. The goal is to develop tailored strategies to mitigate these risks, thereby reducing the occurrence of associated disorders and their potential impact on this specific group of military personnel. 

The finding highlighted the highest pain intensity in the back region, specifically during the tasks of the commander, gunner, and driver. However, the loader showed the highest intensity in the knees. These results appear to be consistent with those from previous studies, showing a higher prevalence of spinal and lower limb pain in military personnel [6,7]. Concerning each task, the commander reported higher pain intensity in the knees and back, consistent with a study among German MBT commanders [18]. The authors attributed these results to vibrations during standing. Although we did not collect vibration data and the data were collected with the MBT stopped, we can speculate that one of the reasons for these results is the presence of vibrations during the movement of the tank. According to Nissen et al. [17], the gunner is at risk of developing neck problems, which in our study is reflected in the highest pain intensity in the back and neck during this task. A possible explanation could be isometric muscle activity over a longer period to maintain the same posture. The loader has a higher intensity of pain in the back area, which is probably related to handling heavy ammunition in the tight interior of the tank.

When looking at the risk analysis and the posture of the individual military personnel, it is noticeable that this posture involves bending and twisting movements. These postures place ever greater demands on the body segments and muscles involved, which ultimately leads to greater wear and tear on the joints and, thus, greater physical overload and greater muscle fatigue.

Regarding the risk assessment, it was expected that the final score for all tasks would be significantly higher, since the most demanding position of each task was chosen for this analysis and REBA is a method that does not depend on specific benchmarks but rather uses position and joint angle in relation to each other to assess the MSKI risk [26]. All four tasks had classifications between medium and high. The higher scoring tasks were the ones that took up the majority of the mission duration. According to Lovalekar et al. when assessing the risk, the time required to carry out the activity must also be taken into account [11]. Therefore, after creating the normalized risk score (SRn) indicator, which takes into account the time spent on each task, it was possible to identify the most critical tasks of each crew member that contribute to a higher risk of MSKI [5]. When comparing the REBA baseline scores with the normalized risk score, the risk map changes. Most tasks are low scoring, while high- and very-high-scoring tasks are expected to take place over long periods.

Implementing tailored strategies to mitigate these risks is critical to reducing the risk of MBT crews developing MSKIs. For example, since the risk factors are different than those for other units, implementing a tailored training model for this crew can protect these military personnel. Small adjustments could be made inside the tank to improve the posture of the soldiers. These changes and adjustments should be carried out carefully so as not to endanger the safety of the crew.

This study has some noteworthy limitations. The data collection was very time-consuming, and the resources required and used were high. It was not possible to carry out a kinematic analysis while the Leopard 2 A6 MBT was in motion, making it impossible to study the influence of other factors such as vibration and noise. Whole-body vibration during operation in rough terrain is a condition for tank crews [27] and is critical in musculoskeletal studies. Future studies should consider these factors for a more comprehensive analysis of all risk factors contributing to MSKIs in the military context. Since the space in the Leopard 2 A6 MBT is extremely small, we were unable to video record and then apply REBA, so we had to use the IMUs to obtain the data. The advantage of IMUs is that all postures of military personnel are continuously recorded for two hours. Our study did not take into account psychosocial factors that could influence the perceptions of pain and risk among the tank crew members. Future studies should consider assessments of these factors to provide a more comprehensive understanding of the determinants of musculoskeletal injury risk. Furthermore, since this is a pilot study and the sample size is small, the results cannot be generalized and future studies with a representative sample are required.

Overall, this study adds valuable insights to the growing literature on MSKI prevention in military personnel by providing a comprehensive assessment of risk factors for specific crew roles. The high prevalence of MSKI is consistent with previous research [6,7,17] and highlights the need for proactive prevention strategies beyond acute traumatic injuries. The focus on cumulative microtraumatic mechanisms highlights the importance of recognizing and addressing the long-term effects of repetitive tasks on musculoskeletal health. 

## Figures and Tables

**Figure 1 sensors-24-04527-f001:**
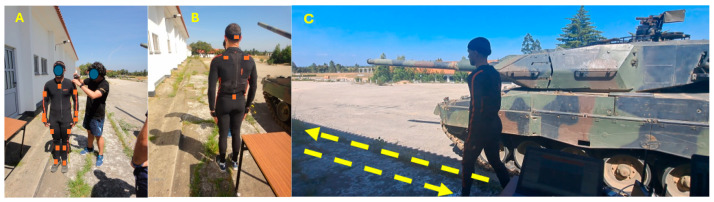
(**A**,**B**) Participant in the “n-pose” with the IMU suit with 17 sensors, represented by orange rectangles; (**B**) researcher who collected the anthropometric measurements to insert them into the MVN software; (**C**) participant performing dynamic calibration (following the yellow arrows) of the IMU system to ensure alignment of sensors to body segments, Leopard 2 A6 Main Battle Tank, and a personal computer with data acquisition using MVN Analyse software.

**Figure 2 sensors-24-04527-f002:**
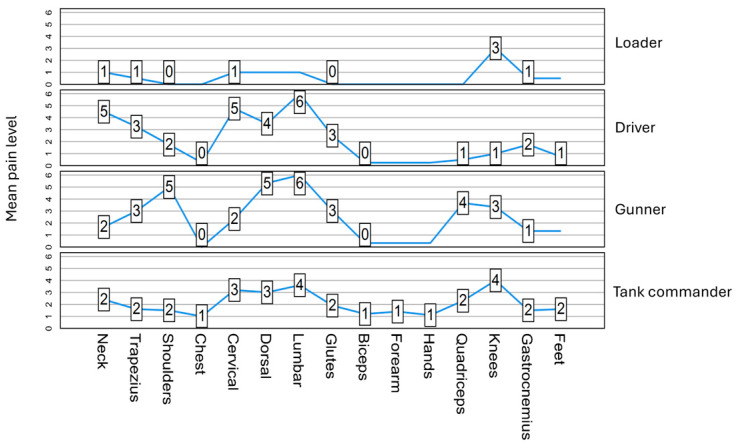
Pain intensity per body segment for each task while performing the task.

**Figure 3 sensors-24-04527-f003:**
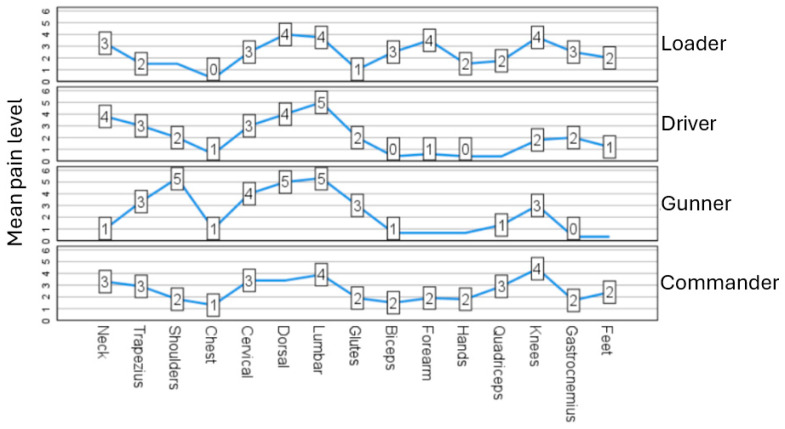
Pain intensity per body segment after task performance.

**Figure 4 sensors-24-04527-f004:**
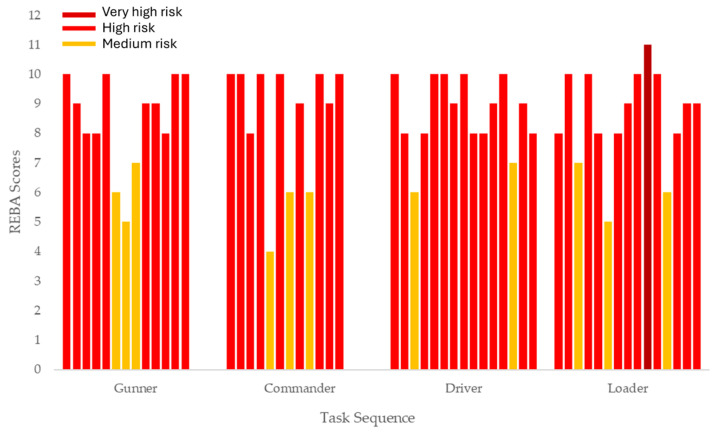
Final REBA score for all tasks: gunner, tank commander, driver, and loader. Each bar corresponds to a step in the task sequence.

**Table 1 sensors-24-04527-t001:** Description of the sequence of tasks performed by the tank commander in the MBT.

Tank Commander’s Tasks		
No.	Description	Time (min)
1	Climb into the tank with the “three-legged rule”	1
2	Entrance through the tank commander’s hatch	1
3	Placement in the tank commander´s position	1
4	Observation from outside the hatch	40
5	Retracting the tank commander’s seat	1
6	Tank commander´s hatch lock	1
7	Observation through the block visions and through the tank commander´s monitor	70
8	Opening the tank commander´s hatch	1
9	Positioning the seat for the observation position outside the hatch	2
10	Exit from inside the tank through the tank commander’s hatch	1
11	Stepping down from the turret to the hull	0.5
12	Stepping down the tank	0.5

**Table 2 sensors-24-04527-t002:** Description of the sequence tasks performed by the gunner in the MBT.

Gunner’s Tasks		
No.	Description	Time (min)
1	Climbing into the tank with the “three-legged rule”	1
2	Entrance through the tank commander’s hatch	1
3	Placement in the gunner position	1
4	Back support placement	1
5	Operating the computer control panel and display and the gunner´s control and display panel	1
6	Diopter dial and of the face support adjustment	1
7	Firing sequence under the control of gunner	108
8	Removing the back support	1
9	Lowering the gunner´s seat	1
10	Getting out of the gunner position	1
11	Exiting from inside the tank through the tank commander’s hatch	1
12	Stepping down from the turret to the hull	1
13	Stepping down the tank	1

**Table 3 sensors-24-04527-t003:** Description of the sequence tasks performed by the driver in the MBT.

Driver’s Tasks		
No.	Description	Time (min)
1	Climbing into the tank with the “three-legged rule”	1
2	Entrance through the tank commander’s hatch	0.5
3	Passing the gunner’s compartment	0.5
4	Getting into the driver´s compartment	0.5
5	Placement in the driver´s position	0.5
6	Checking extinguishers for fire	1
7	Driver control box checks	3
8	Driving	107
9	Exiting from inside the tank through the tank commander’s hatch	2
10	Stepping down from the turret to the hull	0.5
11	Stepping down the tank	0.5
**Sequence of tasks in case of exit through the emergency hatch**		
12	Removing the back support	1
13	Opening the driver´s emergency hatch	1
14	Exiting from inside the tank through the emergency hatch	0.5

**Table 4 sensors-24-04527-t004:** Description of the sequence tasks performed by the loader in the MBT.

Loader’s Tasks		
No.	Description	Time (min)
1	Climbing into the tank with the “three-legged rule”	1
2	Installing the anti-aircraft gun in the holder	2
3	Entrance through the loader’s hatch	1
4	Checking the main gun	2
5	Radio installation	2
6	Adjusting the loader´s panel	14
7	Opening the breech ring	16
8	Opening the ammunition compartment (bunker)	16
9	Taking out a cartridge from the bunker	16
10	Placing the cartridge into the cartridge chamber	16
11	Loading the main gun by pushing the cartridge completely into the cartridge chamber	16
12	Calling out and assuming the position of “Loader ready”	16
13	Exiting from inside the tank through the loader’s hatch	1
14	Stepping down from the turret to the hull	0.5
15	Stepping down the tank	0.5

**Table 5 sensors-24-04527-t005:** Normalized risk score levels.

Score SRn_i_	Risk Level	Action
1–7	Negligible	Not necessary
8–20	Low	Possibly necessary
21–47	Medium	Necessary
48–67	High	Soon necessary
68–100	Very high	Immediately necessary

**Table 6 sensors-24-04527-t006:** Characterization of the participants who answered the questionnaires.

Task	Gender		Height (cm)			Mass (kg)			Age (Years)		
	Male	Female	Min.	Max.	Mean ± SD	Min.	Max.	Mean ± SD	Min.	Max.	Mean ± SD
Tank commander	31	6	157	191	174.95 ± 7.76	53	103	74.32 ± 11.24	22	52	33.65 ± 6.71
Gunner	7	0	162	183	173.00 ± 8.12	55	110	74.29 ± 17.55	19	32	23.14 ± 4.45
Driver	7	0	172	180	176.29 ± 2.98	64	80	73.54 ± 5.78	19	24	21.57 ± 1.62
Loader	5	1	165	185	177.17 ± 8.61	60	84	74.67 ± 9.61	20	33	25.33 ± 5.39

SD—standard deviation.

**Table 7 sensors-24-04527-t007:** Characterization of the four male participants who performed the kinematic analysis.

Task	Height (cm)	Mass (kg)	Age (Years)	Seniority (Years)
Tank commander	170	75	26	2
Gunner	183	80	21	1
Driver	171	77	22	2
Loader	172	70	19	1

**Table 8 sensors-24-04527-t008:** REBA scores of the four military personnel belonging to the Leopard 2 A6 command centre (CC) crew. The colors correspond to the different REBA risk levels: green for negligible; yellow for low, orange for high and red for very high.

Gunner					Tank Commander				
Task	Score REBA	Time (min)	SR	SRn	Task	Score REBA	Time (min)	SR	SRn
1	10	1	8	2	1	10	1	8	2
2	9	1	8	2	2	10	1	8	2
3	8	1	7	1	3	8	1	7	2
4	8	1	7	1	4	10	40	333	95
5	10	1	8	2	5	4	1	3	1
6	6	1	5	1	6	10	1	8	2
7	5	108	450	100	7	6	70	350	100
8	7	1	6	1	8	9	1	8	2
9	9	1	8	2	9	6	2	10	3
10	9	1	8	2	10	10	1	8	2
11	8	1	7	1	11	9	0.5	4	1
12	10	1	8	2	12	10	0.5	4	1
13	10	1	8	2					
**Driver**					**Loader**				
**Task**	**Score REBA**	**Time** **(min)**	**SR**	**SRn**	**Task**	**Score REBA**	**Time (min)**	**SR**	**SRn**
1	10	1	8	1	1	8	1	7	5
2	8	0.5	3	1	2	10	2	17	11
3	6	0.5	3	1	3	7	1	6	4
4	8	0.5	3	1	4	10	2	17	11
5	10	0.5	4	1	5	8	2	13	9
6	10	1	8	1	6	5	14	58	40
7	9	3	23	3	7	8	16	107	73
8	10	107	892	100	8	9	16	120	82
9	8	2	13	1	9	10	16	133	91
10	8	0.5	3	1	10	11	16	147	100
11	9	0.5	4	1	11	10	16	133	91
12	10	1	8	1	12	6	16	80	55
13	7	1	6	1	13	8	1	7	5
14	9	0.5	4	1	14	9	0.5	4	3
15	8	0.5	3	1	15	9	0.5	4	3

## Data Availability

Data available on request due to restrictions (Military restrictions).

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
