# Peer review of "Risk Factors Associated with Musculoskeletal Injuries within the Crew of the Leopard 2 A6 Main Battle Tank Using Inertial Movement Unit Sensors: A Pilot Study"

_sensors, 2024, doi:10.3390/s24144527_

Round 1

Reviewer 1 Report

Comments and Suggestions for Authors

1. The implementation details for reproducing the study are inadequate, particularly concerning the calibration and placement of IMU sensors. Can you provide more detailed implementation steps for the calibration and placement of IMU sensors to ensure reproducibility?

2. The evaluation and ablation studies for the proposed methods are limited, making it difficult to assess the robustness of the findings. For example, what are the specific criteria used for selecting the four soldiers for the kinematic analysis, and how do you justify the sample size?

3. There is a lack of thorough comparison with widely-known baselines in the field of military ergonomics and MSKI prevention. Can you include more detailed comparisons with existing studies and baseline data on MSKI in military personnel?

4. The theoretical analysis does not sufficiently address the potential impact of vibration and noise, which are critical factors in a tank environment.  How do you account for the potential effects of vibration and noise in your analysis, and can these be incorporated into future studies?

5. The exposition lacks clarity in certain sections, particularly in the description of the statistical methods and their justification. Can you clarify the statistical methods used for data analysis and provide a rationale for their selection?

6. The study does not discuss the potential psychological and cognitive factors that might influence the perception of pain and risk among the tank crew members. How do you plan to address the psychological and cognitive factors that might influence the perception of pain and risk?

7. There is a need for more detailed recommendations on how the proposed mitigation measures can be practically implemented within the military context.

Comments on the Quality of English Language

The quality of the English language in the manuscript is generally good, but there are a few areas where improvements can be made for better clarity and readability

Author Response

We thank the reviewers for the helpful comments. All changes to our manuscript are highlighted by using the track changes mode tool. We have revised our manuscript according to these comments, which we believe have further improved the quality of the manuscript.

Reviewer 1

Reviewer Point #1:  The implementation details for reproducing the study are inadequate, particularly concerning the calibration and placement of IMU sensors. Can you provide more detailed implementation steps for the calibration and placement of IMU sensors to ensure reproducibility?  

Author response #1: We are thankful for your comments. In sections 2.5. Instrumentation and 2.6. Kinematic Data Processing, we introduced more detail about the placement of the IMU sensors in the Lycra suit and the calibration process. We also added a figure to clearly understand the location of the sensors and the calibration process.

Reviewer Point #2: The evaluation and ablation studies for the proposed methods are limited, making it difficult to assess the robustness of the findings. For example, what are the specific criteria used for selecting the four soldiers for the kinematic analysis, and how do you justify the sample size?

Author response #2:  To address this concern we added we added the inclusion and exclusion criteria in section 2.2. Participants: “This study included the population of 57 military personnel who are currently serving or have previously served in a MBT Leopard 2 A6 crew by asking to respond to a questionnaire. Inclusion criteria were both gender, to have experience at least of one year in the military service, military workers that take or took part of Tank Commander, the Gunner, the Driver and the Loader tasks. Exclusion criteria was to have a musculoskeletal disorder one year previous to data collection. A subgroup of four military personnel was randomly selected to perform the main daily tasks, in order collect quantitative data on biomechanical work exposure. Before starting the data collection, all participants were informed about the study, its objectives, and procedures, and an informed consent form was read and signed.”

Reviewer Point #3: There is a lack of thorough comparison with widely-known baselines in the field of military ergonomics and MSKI prevention. Can you include more detailed comparisons with existing studies and baseline data on MSKI in military personnel?  

Author response #3:  Thank you for your valuable feedback. To address your comment, we have made a more detailed comparison with the study by Nissen at al. 2009 and added another reference: Sandmann and Gurske, 2003. Both studies analyzed the prevalence of musculoskeletal pain in battle tank crews. We added this text to the Introduction section: A study comparing the Danish army MBT crew with personnel from other unit types found that the MSKI pattern did not differ significantly between different units [17]. Within the MBT crew, working as a gunner for less than two years was associated with an increased risk of reporting neck pain, while working as a loader was linked to a higher risk of shoulder pain [17]. Another study among German MBT commanders revealed significant back and knee pain after a 320km road deployment [18].”

The two references added:

  1. Nissen, R.; Guldager, B.; Gyntelberg, F. Musculoskeletal Disorders in Main Battle Tank Personnel. Mil Med 2009, 174, 952–957, doi:10.7205/MILMED-D-03-5508.
  2. Sandmann, J.; Gurske, S. Health Disturbances of German Battle Tank Officers: Results of an Interview with 64 Commanding Officers of a Tank Battalion. Mil Med 2003, 168, 715–721.

Reviewer Point #4: The theoretical analysis does not sufficiently address the potential impact of vibration and noise, which are critical factors in a tank environment. How do you account for the potential effects of vibration and noise in your analysis, and can these be incorporated into future studies?

Author response #4: Thank you for your insightful comment. We agree that vibration and noise are significant factors in the tank environment and should be considered in the analysis. We added a sentence to the limitation part in the Discussion section: “Whole-body vibration during operation in rough terrain is a condition for tank crews [32] and is critical in musculoskeletal studies. Future studies should consider these factors for a more comprehensive analysis of all risk factors contributing to MSKI in the military context.”

Reviewer Point #5: The exposition lacks clarity in certain sections, particularly in the description of the statistical methods and their justification. Can you clarify the statistical methods used for data analysis and provide a rationale for their selection?

Author response #5:  We have added more information about the data analysis to the 2.8 Statistical Analysis section: “The data obtained from the REBA assessment and pain intensity, was tested for normality (Shapiro Wilk test) and homogeneity of variances (Levene’s test). If both pre-requirements were not observed, a One-way Anova, with Welsh correlation was conducted in order to analyse differences among military tasks. Post hoc Tamhane method was used to determine which of REBA scores differ among MBT tasks.  After, a Spearman correlation between the REBA scores and pain intensity was conducted in order to analyse the impact of biomechanics exposure on the experienced symptoms. Again, if the normality of variables will not observe, we use the Spearman correlation.”

Reviewer Point #6: The study does not discuss the potential psychological and cognitive factors that might influence the perception of pain and risk among the tank crew members. How do you plan to address the psychological and cognitive factors that might influence the perception of pain and risk?

Author response #6: Thank you for your insightful comment. We recognize the importance of psychological and cognitive factors in the perception of pain. To address this, we have added the following sentences to the discussion section: “Our study did not account psychosocial factors that could influence perceptions of pain and risk in tank crew members. Future studies should consider assessments of these factors to provide a more comprehensive understanding of the determinants of musculoskeletal injury risk.”

Reviewer Point #7: There is a need for more detailed recommendations on how the proposed mitigation measures can be practically implemented within the military context.

Author response #7:  We are thankful for your comment, which we addressed in the discussion section: “Implementing tailored strategies to mitigate these risks is critical to reduce the risk of MBT crews developing MSKI. For example, implementing a tailored training model for this crew, as the risk factors are different than other units, that can protect these military personnel. Small adjustments could be made inside the tank to improve the posture of the soldiers. These changes and adaptations should be caried out carefully so as not to endanger the safety of the crew.”

Reviewer Point #8: The quality of the English language in the manuscript is generally good, but there are a few areas where improvements can be made for better clarity and readability.

Author response #8:  Thank you for your feedback on the quality of the English language in our manuscript. We revise the manuscript to improve the quality of English.

Reviewer 2 Report

Comments and Suggestions for Authors

The aim of this study was to determine the degree of musculoskeletal risk among military personnel of a Leopard 2 A6 main battle tank (MBT) crew in the performance of their tasks and to identify associated risk factors. The ultimate goal is to develop mitigation measures tailored to this specific group to reduce the incidence of musculoskeletal disorders and their possible consequences.

Comments in pdf.

Comments on the Quality of English Language

Extensive proofreading is needed.

Author Response

We thank the reviewers for the helpful comments. All changes to our manuscript are highlighted by using the track changes mode tool. We have revised our manuscript according to these comments, which we believe have further improved the quality of the manuscript.

Reviewer 2

The aim of this study was to determine the degree of musculoskeletal risk among military personnel of a Leopard 2 A6 main battle tank (MBT) crew in the performance of their tasks and to identify associated risk factors. The ultimate goal is to develop mitigation measures tailored to this specific group to reduce the incidence of musculoskeletal disorders and their possible consequences.

Comments in pdf.

Reviewer Point #1:  Break this sentence into 2 sentences.

Author response #1: We thank you for the suggestion and will amend it accordingly: “Musculoskeletal injuries (MSKI) are the leading cause of medical discharge and medical downgrade in the armed forces. They have a major impact on deployability and combat effectiveness and are endemic in the military population [1].”

Reviewer Point #2: You mean deployability here?

Author response #2:  We thank the reviewer for catching this typo and change it to deployability.

Reviewer Point #3: Consider breaking down complex sentences for clarity.

Author response #3:  We agree with the suggestion and break the sentence in two for more clarity: “Previous military studies have suggested that 30-70% of military injuries are musculoskeletal overuse injuries (MSIO) [2,3]. Nonetheless, most of these studies and national surveillance systems only include diagnosis of acute traumatic injuries [4,5].”

Reviewer Point #4: Is this the correct word?

Author response #4:  What do we mean by national surveillance system is: the surveillance system for collection, analysis, and sharing of health data, resources, and information about policies and standards at the local, state, and national levels.

Reviewer Point #5: and

Author response #5:  We change it accordingly.

Reviewer Point #6: these

Author response #6: We change it accordingly.

Reviewer Point #7: often

Author response #7: We change it accordingly.

Reviewer Point #8: accounts

Author response #8: We thank the reviewer for catching this typo and change it to accounts.

Reviewer Point #9: Paragraph needs restructuring. This cannot be a single paragraph break this into 3-4 paragraphs.

Author response #9: We restructured the entire paragraph and divided it into small paragraphs to make the introduction clearer and more readable.

Reviewer Point #10: ?

Author response #10: Thank you for noticing this typo. The correct word is previous and we change it accordingly.

Reviewer Point #11: What do you mean here?

Author response #11: By uncontrolled variables we mean any factor that cannot be controlled by the researcher, for example weather conditions or terrain conditions, in the case of the military field. We have added these examples to the manuscript to make it clearer.

Reviewer Point #12: This sentence is not clear.

Author response #12: We change the sentence to clarify: However, due to the unique nature of military activities, quantitative measurements should be prioritized, as normative values from other occupational settings may not accurately reflect the conditions and exposures experienced by military personnel.”

Reviewer Point #13: So what? What is the aim of the mentioned studies? What did they find?

Author response #13: We have added more details to the study mentioned and added another study developed on MBT crews: There is little research addressing the incidence of MSKI among MBT crews. A study comparing the Danish army MBT crew with personnel from other unit types found that the MSKI pattern did not differ significantly between different units [15]. Within the MBT crew, working as a gunner for less than two years was associated with an increased risk of reporting neck pain, while working as a loader was linked to a higher risk of shoulder pain [17]. Another study among German MBT commanders revealed significant back and knee pain after a 320km road deployment [18].”

Reviewer Point #14: Pease provide more details about the study design.

Author response #14: More information has been added in section 2.1. Study design: “This was a single experimental study (simulation) conducted with military personnel of the Portuguese Army.”

Reviewer Point #15: I cannot understand what you mean here.

Author response #15: The participants in the study were all the military personnel who are part of the MBT crew or have served in this crew in the paste. The latter are still active military members. To clarify we change the sentence in section 2.2 Participants: “This study included the population of 57 military personnel who are currently serving or have previously served in a MBT Leopard 2 A6 crew by asking to respond to a questionnaire. Inclusion criteria were: both gender, to have experience at least of one year in the military service, military workers that take or took part of Tank Commander, the Gunner, the Driver and the Loader tasks. Exclusion criteria was to have a musculoskeletal disorder one year previous to data collection.”

Reviewer Point #16: This is confusing. More details are needed.

Author response #16: We added more information about the selection of the military personnel to the experimental study: “A subgroup of four military personnel was randomly selected to perform the main daily tasks, in order collect quantitative data on biomechanical work exposure. Before starting the data collection, all participants were informed about the study, its objectives, and procedures, and an informed consent form was read and signed.”

Reviewer Point #17: 0.5

Author response #17: These changes have been made throughout the manuscript.

Reviewer Point #18: Correct all to 0.5

Author response #18: These changes have been made throughout the manuscript.

Reviewer Point #19: What is the name of the scale?

Author response #19: We used a numeric rating scale since is a pain screening tool, commonly used to assess pain severity using a 0–10 scale, with zero meaning “no pain” and 10 meaning “the worst pain imaginable”. The reference to numeric rating scale is in the manuscript.

Reviewer Point #20: Both sides?

Author response #20: We have extended the text added figure 1 to illustrate were the sensors where placed the sensors were placed. You can see this changes in the section 2.5. Instrumentation: “Whole-body kinematic data were recorded at 240 Hz with 17 IMU sensors (Xsens MVN Technologies, Enschede, NL) [21], using the Xsens MVN Analyse software (version 2019.2). The Xsens MVNV Link system consists of 17 IMUs (36 x 24 x 10 mm, 10 g), each containing 3D gyroscopes, 3D accelerometers and a magnetometer, connected to a case and a battery. These sensors were placed inside a Lycra suit after being put on by the participant, specifically, over the feet, shanks, thighs, pelvis, sternum, head, scapulae, upper arms, forearms and hands (Figure 1 A and B). Follow the manufacturer’s in-structions and if possible, position over the bone to reduce soft tissue artifacts [22]. In addition, the body, battery pack and cables were placed in the Lycra suit [23,24]. After all sensors were placed, the following anthropometric measurements were collected from each participant to scale their avatar model: standing height, shoe length, arm span, ankle, knee, hip and shoulder heights, hip and shoulder width, and shoe sole height [23,25] (Figure 1A). A calibration procedure was then carried out to align the sensors to the respective segment [26]. This calibration was performed by having the participants stand in a neutral “n-pose” position with the palm facing medially, then walk forward a few meters, return to the starting position and stand in “n-pose” again [27] (Figure 1C). Following the calibration procedures, each military personnel individually performed a set of tasks inherent to their role in the MBT (Table 1, 2, 3, 4 and Appendix 1).”

Reviewer Point #21: Please provide more information? Was a body model used?

Author response #21: We have included figure 1 to illustrate where the sensors were placed in the instrumentation section. A biomechanical model was created based on each participant’s anthropometric dimensions and Xsens calibration procedures. We described in

the 2.5. instrumentation and 2.6. kinematic and processing section.

Reviewer Point #22: What is this?

Author response #22: This is a static calibration position, and now it’s possible to visualize it

in the figure 1 A and 1B.

Reviewer Point #23: Sentence does not make sense.

Author response #23: We rephrase the sentence to make it clearer: “To determine the biomechanics risk associated to the performance of each task, the joint angle data were determined and used to obtain the Rapid Entire Body Assessment (REBA) scores method [31].”

Reviewer Point #24: Did you use Visual 3D to get the angles?

Author response #24: Yes, we use the Visual 3D software to process the kinematic data. You

can see it in the kinematic and data processing section.

Reviewer Point #25: determined

Author response #25: We change it accordingly.

Reviewer Point #26: What is this?

Author response #26: This is the abbreviation for work-related musculoskeletal disorders.

We change it to the abbreviations used through the manuscript: MSKI.

Reviewer Point #27: previous

Author response #27: We change it accordingly.

Reviewer Point #28: ????

Author response #28: We change the term to military personnel.

Reviewer Point #29: what about muscle injuries?

Author response #29: We rephrase the sentence to incorporate your suggestion: These postures place increasing demands on the body segments and muscles involved. This leads to greater wear and tear on the joints, resulting in increased physical overload and muscle fatigue, which can ultimately lead to a musculoskeletal injury (MSKI).”

Reviewer Point #30: what about muscle strains?

Author response #30: We rephrase the sentence to include your suggestion: These postures place increasing demands on the body segments and muscles involved. This leads to greater wear and tear on the joints, resulting in increased physical overload and muscle fatigue, which can ultimately lead to a musculoskeletal injury (MSKI)."

Reviewer Point #31: Was this included in the methods?

Author response #31: Tables 1 to 4, list the order of tasks of the driver, tank commander, loader and gunner and the time required to perform each. In section 2.4. MBT tasks – experimental protocol we mention the following: “The sequence of tasks was analysed with based on a two-hour deployment and allowed tasks to be divided according to execution time.”

Reviewer Point #32: This should be expanded to discuss how these limitations might have influenced the results.

Author response #32: We expanded the discussion about the limitations and how they might affect the results: Whole-body vibration during operation in rough terrain is a condition for tank crews [32] and is critical in musculoskeletal studies. Future studies should consider these factors for a more comprehensive analysis of all risk factors contributing to MSKI in the military context. Since the space inside the MBT Leopard 2 A6 is extremely small, we were unable to video record and then apply REBA, so we had to use the IMUs to obtain the data. The advantage of IMUs is that all postures of military personnel are continuously recorded for two hours.”

Reviewer Point #33: What about future research?

Author response #33: We outlined future research directions in the discussion:It is known that exposure to whole-body vibration while driving in rough terrain is an operating condition for tank crews [32] and is critical in musculoskeletal studies. Future studies should consider these factors. Since the space inside the MBT Leopard 2 A6 is extremely small, we were unable to video record and then apply REBA, so we had to use the IMUs to obtain the data. Our study did not account psychosocial factors that could influence perceptions of pain and risk in tank crew members. Future studies should consider assessments of these factors to provide a more comprehensive understanding of the determinants of musculoskeletal injury risk.”

Reviewer 3 Report

Comments and Suggestions for Authors

Dear authors,

I carefully analyzed your article. The idea is an interesting one, and the field shows the usefulness of possible discoveries. But, until you can extract a clear and useful conclusion, you must prepare a scientific article more thoroughly and strictly.

I recognize the originality of your ideas, given the field in which you wanted to bring news.

Below I give you my observations.

·       The standard methodology is not respected, details that would bring clarity and understanding are missing, the design of the study being inadequate.

·       The familiarization, test session, including the experimental protocol cannot be understood exactly, because no details are given.

·       and many details are missing.

·       The little data about the subjects, the insufficient criteria for their inclusion and exclusion, the small number of subjects?? (ok, of course the field from which the subjects belong, but their number is still small), the size of the sample ???? (does not provide fidelity and depth to the statistical analysis), the lack of details about the results, etc., etc., all do not allow the possibility that this study will be later reproduced by other researchers and does not bring strong arguments for it to be a solid study.

·       No differences or comparisons are presented between the positions of driver, commander, loader, gunner.....

·       The statistics section does not present information that would provide sufficient data about the determined statistical tests.

·       The results present data and tables without statistical analysis (p-values) and it would be necessary to apply a correlation between the factors and their level of musculoskeletal risk.

·       The data from the REBA score (for the 4 types of subjects) should be added in the same plot

·       The discussion section is insufficient.

I hope my observations will help you for the future

Author Response

We thank the reviewers for the helpful comments. All changes to our manuscript are highlighted by using the track changes mode tool. We have revised our manuscript according to these comments, which we believe have further improved the quality of the manuscript.

Reviewer 3

I carefully analyzed your article. The idea is an interesting one, and the field shows the usefulness of possible discoveries. But, until you can extract a clear and useful conclusion, you must prepare a scientific article more thoroughly and strictly.

I recognize the originality of your ideas, given the field in which you wanted to bring news.

Below I give you my observations.

Reviewer Point #1:  The standard methodology is not respected, details that would bring clarity and understanding are missing, the design of the study being inadequate.

Author response #1: To clarify the study design we have information in section 2.1. Study design: “This was a single experimental study (simulation) conducted with military personnel of the Portuguese Army.”

Reviewer Point #2: The familiarization, test session, including the experimental protocol cannot be understood exactly, because no details are given, and many details are missing.

Author response #2:  We are grateful for your comment. We explained the familiarization test, which consisted of the sequence of tasks in the MBT described in Tables 1, 2, 3 and 4, for each crew member. We have also added the experimental protocol presentation in Appendix 1.

Reviewer Point #3: The little data about the subjects, the insufficient criteria for their inclusion and exclusion, the small number of subjects?? (ok, of course the field from which the subjects belong, but their number is still small), the size of the sample ???? (does not provide fidelity and depth to the statistical analysis), the lack of details about the results, etc., etc., all do not allow the possibility that this study will be later reproduced by other researchers and does not bring strong arguments for it to be a solid study.

Author response #3:  To address this concern we added we added the inclusion and exclusion criteria in section 2.2. Participants: “This study included the population of 57 military personnel who are currently serving or have previously served in a MBT Leopard 2 A6 crew by asking to respond to a questionnaire. Inclusion criteria were both gender, to have experience at least of one year in the military service, military workers that take or took part of Tank Commander, the Gunner, the Driver and the Loader tasks. Exclusion criteria was to have a musculoskeletal disorder one year previous to data collection. A subgroup of four military personnel was randomly selected to perform the main daily tasks, in order collect quantitative data on biomechanical work exposure. Before starting the data collection, all participants were informed about the study, its objectives, and procedures, and an informed consent form was read and signed.”

Reviewer Point #4: No differences or comparisons are presented between the positions of driver, commander, loader, gunner.....

Author response #4:  Thank you for your insightful comment. We welcome your suggestion to include a comparison between the different positions within the tank crew. To address this, we added Figure 3, which contains the REBA results for all tasks, allowing for easier comparison between tasks. We also review the Results section to add a more detailed comparison between the four tasks.

Reviewer Point #5: The statistics section does not present information that would provide sufficient data about the determined statistical tests.

Author response #5:  More detailed information was added to the 2.8 Statistical Analysis section: “The data obtained from the REBA assessment and pain intensity, was tested for normality (Shapiro Wilk test) and homogeneity of variances (Levene’s test). If both pre-requirements were not observed, a One-way Anova, with Welsh correlation was conducted in order to analyse differences among military tasks. Post hoc Tamhane method was used to determine which of REBA scores differ among MBT tasks.  After, a Spearman correlation between the REBA scores and pain intensity was conducted in order to analyse the impact of biomechanics exposure on the experienced symptoms. Again, if the normality of variables will not observe, we use the Spearman correlation.”

Reviewer Point #6: The results present data and tables without statistical analysis (p-values) and it would be necessary to apply a correlation between the factors and their level of musculoskeletal risk.

Author response #6: To address this issue, we added this sentence to 3.3. Kinematic data and assessment risk analysis:Regarding the analysis of biomechanics exposure, there are differences between the REBA normalized among Tank Commander and Gunner tasks (p<0.001), as well as among Gunner and driver tasks (p<0.001) and Loader tasks (p<0.001). However, there were no statistical differences among all tasks when considering the crude REBA score. This results calls the attention the need to consider the time exposure in parallel to the biomechanics risk analysis.

Additionally, these differences were not reflecting in the correlation between REBA normalized and the pain intensity in the different body segments, either during the tasks and after.”

Reviewer Point #7: The data from the REBA score (for the 4 types of subjects) should be added in the same plot.

Author response #7:  Thank you for your valuable suggestion. We created a combined plot containing the REBA scores for all four subjects. This plot provides a clearer comparison and better visualization of the differences and similarities in musculoskeletal risk between the different subjects. The revised figure can be found in the Results section, Figure 3. This combined plot allows for easier interpretation of the data and highlights the different levels of risk across different tasks and roles within the crew. The images depicting the pose analysis for each sequence step have been moved to Appendix I. The text in the Results section on the REBA figures has also been changed accordingly. All changes are marked with the track change tool.

Reviewer Point #8: The discussion section is insufficient.

Author response #8: Thank you for your valuable feedback. We have revised to improve the discussion of the manuscript, which we marked using the track change tool.

Round 2

Reviewer 1 Report

Comments and Suggestions for Authors

The author's reply answered my question. I have no further questions.

Author Response

Dear Reviewer,

Thank you for your positive feedback. We are pleased that our response satisfactorily answered your questions. We thank you for your time and consideration in reviewing our manuscript.

Reviewer 2 Report

Comments and Suggestions for Authors

Dear authors,

Please see the attached document with my comments.

Comments on the Quality of English Language

Extensive proofreading needed.

Author Response

Manuscript ID: sensors-3050632

Dear Reviewer,

We thank you for the time and effort you have given to provide your feedback. We are grateful for the insightful comments and valuable suggestions that significantly improved our paper. We considered your suggestions into account and highlighted these changes using the change tracking tool. The entire manuscript underwent a full English revision. Below is a point-by-point response to your comments and concerns.

Reviewer Point #1:  I will call this a pilot study.

Author response #1: We agree with the reviewers’ suggestion and change the title accordingly: Risk Factors Associated with Musculoskeletal Injuries Within 2 the Crew of the Leopard 2 A6 Main Battle Tank Using IMU 3 Sensors: a pilot study

Reviewer Point #2: Space

Author response #2: Thank you for catching this typo.

Reviewer Point #3: This is not academic writing. What do you mean as you move up the scale? Please rephrase.

Author response #3:  We rephrase the sentence: “As we improve the quality of results and accuracy of the analysis, the costs of time and expertise required to analyze the data also increases.”

Reviewer Point #4: prioritised maybe?

Author response #4: We accepted the suggestion and changed it to prioritised.

Reviewer Point #5: significantly differ

Author response #5:  We accepted the suggestion and modified it.

Reviewer Point #6: I suggest starting a new paragraph from here.

Author response #6: We have started a new paragraph, as suggested.

Reviewer Point #7: Portuguese Army military personnel

Author response #7:  Thank you for catching this typo.

Reviewer Point #8: Please rephrase sentence makes no sense here.

Author response #8: The sentence was rephrased: “The population of 57 military personnel were asked to respond to a questionnaire “

Reviewer Point #9: genders

Author response #9: Thank you for catching this typo.

Reviewer Point #10: How many? You indicate 4 afterwards but why only 4? Maybe you may want to call this a pilot study? With 4 people you cannot reach conclusions.

Author response #10: We randomly selected 4 military personnel from the population (n=57), corresponding to the number of tasks that we assessed using direct measures (Xsens). As you suggested, we referred to this study as a pilot study, and added this note in the title of the manuscript.

Reviewer Point #11: Is there a name for this scale.

Author response #11: Numeric Pain Rating Scale (NPRS) is the name of the scale. To provide clarification, we have included additional details about the scale and the psychometric properties:Military personnel were asked to report their discomfort and pain in 15 body regions using a Numeric Pain Rating Scale (NPRS) [20]. The NPRS is a self-report instrument used in this study to assess pain status. The scale was presented with numbers from 0 to 10 with the following anchors at each end of the scale: “no pain” and “worst possible pain.”. The NPRS has been shown to be reliable with an intraclass correlation coefficient (ICC) of 0.76 (95% confidence interval: 0.51–0.87) [20]. Participants reported their pain at two different times: during task performance and after task performance.”

Reviewer Point #12: times

Author response #12: We change it accordingly.

Reviewer Point #13: Ensure the sequence of tasks is easy to follow. Consider summarizing repetitive details to improve readability.

Author response #13: We recognize that the sequence of tasks in each crew function is long and that sometimes these tasks are common between them. However, we believe that the order of tasks of each crew function should be presented independently, as shown in Tables 1, 2, 3, and 4.

Reviewer Point #14: What is this here?

Author response #14: We have deleted this typo.

Reviewer Point #15: Both sites?

Author response #15: We added “on both sides” to make the instrumentation procedures clearer to the reader: These sensors were placed, on both the left and right body segments, of a Lycra suit after the participant put in on, over the feet, shanks, thighs, scapulae, upper arms, forearms hands one sensor in the sternum, one in the head and one in the pelvis, (Figure 1A and 1B).”

Reviewer Point #16: What does this stand for?

Author response #16:  This is the abbreviation for work-related musculoskeletal disorders.

We change it to the abbreviations used in the manuscript: MSKI.

Reviewer Point #17: One paragraph all.

Author response #17: We welcome the suggestion and summarize it in one paragraph.

Reviewer Point #18: You mean Pearson correlation here?

Author response #18: We used a Spearman correlation because the variables did not have a normal distribution and it is a nonparametric measure of rank correlation (statistical dependence between the ranks of two variables). This assesses how well the relationship between two variables can be described using a monotonic function [1].

  1. Spearman, C. (January 1904). "The Proof and Measurement of Association between Two Things" (PDF). The American Journal of Psychology. 15 (1): 72–101. doi:2307/1412159. JSTOR 1412159.

Reviewer Point #19: I cannot find any F-values, p-values and correlation values in the results on the graphs or in the tables.

Author response #19: When describing the results, we present the mean difference between the normalized REBA values for the different crew functions and the p-value for the significant results: “Regarding the analysis of biomechanics exposure, there are statistical significance difference between the REBA normalized among Tank Commander and Gunner tasks (mean difference=8.55; p<0.001). As well as among Gunner and driver tasks (mean difference=1.50; p<0.001) and Loader task (mean difference=1.5; p<0.001). However, there were no statistical differences among all tasks when considering the crude REBA score.  These results call the attention to the need to consider the exposure time for each bio-mechanical risk factor, to account for the daily cumulative effect of that exposure when performing the task.

No statistically significant differences were found between the REBA normalized scores and pain intensity in the different body regions, neither during the performance of the tasks nor afterwards.”

Reviewer Point #20: statistical significance difference?

Author response #20: We change it accordingly.

Reviewer Point #21: I cannot understand what you mean here.

Author response #21: We reformulate the sentence: This results calls the attention to the need to consider the exposure time for each biomechanical risk factor, to account for the daily cumulative effect of that exposure when performing the task.”

Reviewer Point #22: Again, I cannot understand what you mean here.

Author response #22: We rephrase the sentence: “No statistically significant differences were found between REBA normalized scores and pain intensity in the different body regions, neither during task performance nor afterwards.”

Reviewer Point #23: Same participant number?

Author response #23: The sample size varies between different studies, but beyond that the results are consistent between different studies and indicate a higher prevalence of pain in the spine and lower limbs.

Reviewer Point #24: What about sample size? Explain the impact of the small sample size on the generalizability of the findings.

Author response #24: We added a sentence in the discussion when referring to future studies: “Furthermore, since this is a pilot study and the sample size is small, the results cannot be generalized and future studies with a representative sample are required. “

Reviewer Point #25: Is this the conclusion part? What about future research?

Explain the need for larger-scale studies to validate the findings and explore additional factors influencing musculoskeletal injuries in military personnel.

Author response #25: When we mention the limitations of the study, we refer to the need for future studies to investigate other known risk factors not considered in this analysis that contribute to the development of musculoskeletal disorders. In addition, we mention the fact that it was a pilot study with only 4 people, which does not allow us to generalize the data but only take an exploratory approach.

Reviewer 3 Report

Comments and Suggestions for Authors

Dear authors,

I have only one request, as a minor review to adapt:
The significant results are few to provide a strong argument for the statistics (see L272-276). Perhaps the authors should develop the interpretation of the statistical results, even those that do not have a p<0.001 significance, but are relevant as ideas of ensemble that can support the differences.

I hope my observations help you

Author Response

Dear reviewer,

Thank you for your feedback. Below you’ll find the response to you latest comment.

Dear authors,

I have only one request, as a minor review to adapt:

The significant results are few to provide a strong argument for the statistics (see L272-276). Perhaps the authors should develop the interpretation of the statistical results, even those that do not have a p<0.001 significance but are relevant as ideas of ensemble that can support the differences.

Author response #1: If we use the REBA crude scores, the tasks had very similar risk values, which may not account for the cumulative effect of daily exposure when performing the task.
